# Prevalence of and factors associated with multimorbidity among 18 101 adults in the South East Asia Community Observatory Health and Demographic Surveillance System in Malaysia: a population-based, cross-sectional study of the MUTUAL consortium

Michelle M C Tan [ORCID],[1,2] A Matthew Prina [ORCID],[1] Graciela Muniz-Terrera [ORCID],[3,4] Devi Mohan [ORCID],[5] Roshidi Ismail,[6] Esubalew Assefa,[7,8] Ana Á M Keinert,[9] Zaid Kassim,[10] Pascale Allotey,[11,12] Daniel Reidpath,[13,14] Tin Tin Su [ORCID] [5,6]

For numbered affiliations see end of article.

**Correspondence to**
Dr Michelle M C Tan;
tan_mun.chieng@kcl.ac.uk, Dr A Matthew Prina;
matthew.prina@kcl.ac.uk and Professor Tin Tin Su;
TinTin.Su@monash.edu

## ABSTRACT

**Objectives** To assess the prevalence and factors associated with multimorbidity in a community-dwelling general adult population on a large Health and Demographic Surveillance System (HDSS) scale.

**Design** Population-based cross-sectional study.

**Setting** South East Asia Community Observatory HDSS site in Malaysia.

**Participants** Of 45 246 participants recruited from 13 431 households, 18 101 eligible adults aged 18–97 years (mean age 47 years, 55.6% female) were included.

**Main outcome measures** The main outcome was prevalence of multimorbidity. Multimorbidity was defined as the coexistence of two or more chronic conditions per individual. A total of 13 chronic diseases were selected and were further classified into 11 medical conditions to account for multimorbidity. The conditions were heart disease, stroke, diabetes mellitus, hypertension, chronic kidney disease, musculoskeletal disorder, obesity, asthma, vision problem, hearing problem and physical mobility problem. Risk factors for multimorbidity were also analysed.

**Results** Of the study cohort, 28.5% people lived with multimorbidity. The individual prevalence of the chronic conditions ranged from 1.0% to 24.7%, with musculoskeletal disorder (24.7%), obesity (20.7%) and hypertension (18.4%) as the most prevalent chronic conditions. The number of chronic conditions increased linearly with age (p<0.001). In the logistic regression model, multimorbidity is associated with female sex (adjusted OR 1.28, 95% CI 1.17 to 1.40, p<0.001), education levels (primary education compared with no education: adjusted OR 0.63, 95% CI 0.53 to 0.74; secondary education: adjusted OR 0.60, 95% CI 0.51 to 0.70; tertiary education: adjusted OR 0.65, 95% CI 0.54 to 0.80; p<0.001) and employment status (working adults compared

## STRENGTHS AND LIMITATIONS OF THIS STUDY

⇒ This study examined the epidemiology of multimorbidity among community-dwelling adults on a large Health and Demographic Surveillance System scale in a multiethnic and rapidly ageing middle-income country.

⇒ Strengths of the study include the use of a population-based dataset with high participation and completion rate that covers individuals of a broad range of ages and those where chronic disease load begins to manifest.

⇒ We have also captured a more comprehensive range of current and chronic conditions of ageing that have not been included in previous studies of multimorbidity.

⇒ Causality cannot be established due to the cross-sectional nature of the chronic condition data.

with retirees: adjusted OR 0.70, 95% CI 0.60 to 0.82, p<0.001), in addition to age (adjusted OR 1.05, 95% CI 1.05 to 1.05, p<0.001).

**Conclusions** The current single-disease services in primary and secondary care should be accompanied by strategies to address complexities associated with multimorbidity, taking into account the factors associated with multimorbidity identified. Future research is needed to identify the most commonly occurring clusters of chronic diseases and their risk factors to develop more efficient and effective multimorbidity prevention and treatment strategies.

## INTRODUCTION

Multimorbidity, as defined by the WHO,[1] refers to the coexistence of two or more

chronic health conditions in a single individual. The rising tide of multimorbidity is a key challenge for both patients and healthcare systems worldwide.[2] Multimorbidity is associated with lower quality of life,[3] impaired functional capacity,[4] psychological distress,[5] increased premature mortality,[5] and greater healthcare service utilisation and medical expenditure.[6]

There are numerous ways to measure multimorbidity, such as disease counts, conditions weighted or combined with other parameters, diagnostic categories, drug use and physiological measures.[7] Despite these various methods, a simple count of chronic diseases is the most popular method of measuring multimorbidity due to its ease of applying. In a recent systematic review and meta-analysis,[8] it was found that the pooled prevalence of multimorbidity is comparable between high-income countries (HICs) (standardised prevalence 37.9%, 95% CI 32.5% to 43.4%; non-standardised prevalence 41.3%, 95% CI 35.2% to 47.4%) and low-income and middle-income countries (LMICs) (standardised prevalence 29.7%, 95% CI 26.4% to 33.0%; non-standardised prevalence 43.5%, 95% CI 38.4% to 48.6%). While multimorbidity affects all nations and age groups, most studies of the prevalence of multimorbidity have focused on older populations in HICs, and data from younger adults, LMICs and those residing in socioeconomically disadvantaged areas are notably lacking.[2] Similarly, factors that may increase the risk of multimorbidity are poorly understood, with ageing not being the only factor associated with high levels of multimorbidity.[9] Consequently, it has not been possible to evaluate and develop patient-centred prevention and treatment strategies targeting multimorbidity and such risk factors. The COVID-19 pandemic further drew attention to the crucial contribution of multimorbidity to the need for sound public health measures and rapid identification of effective medical interventions grounded on risk stratification.[10] To better manage the consequences of such an adverse phenomenon, a better understanding of the prevalence and factors associated with multimorbidity among diverse study populations became essential.

According to the World Bank Classification,[11] Malaysia is a high-middle-income country in Asia, with a diverse population. After neighbouring Singapore, Malaysia is the second most urbanised country in South East Asia.[12] While it is also experiencing a rapid demographic transition, edging towards population ageing and increasing prevalence of non-communicable diseases,[12] there is limited information on multimorbidity of the general population in the real-world community setting. Very few studies have been able to evaluate the epidemiology of multimorbidity in Malaysia—the World Health Survey[13 14] derived from multimorbidity measures that included a restricted number of conditions and relied on relatively old data. The only contextually relevant studies by in-country researchers were either smaller in sample size[15] or included a limited number of chronic conditions (four and seven conditions, respectively) in the older population.[16 17]

This study is a response of the MUltimorbidity ThroUgh cApacity buiLding (MUTUAL) consortium, developed to better understand and address the growing challenge of multimorbidity in LMICs. The aims of this study were, therefore, to fill the evidence gaps by identifying the country-specific prevalence and factors associated with multimorbidity. To the best of our knowledge, this was the first study that examined in detail a community-dwelling general adult population on a large Health and Demographic Surveillance System (HDSS) scale in Malaysia. To underline the complexity of multimorbidity, and to understand the epidemiology in different age groups, we considered a comprehensive range of chronic health conditions and did not restrict our study to the elderly.

## METHODS
### Data source and study population
Data for the snapshot analyses were drawn from the dynamic South East Asia Community Observatory (SEACO) HDSS database established in 2011 in semiurban and rural areas within the Segamat district of Johor state, Malaysia. The SEACO HDSS profile has been published previously,[18] and is updated here. SEACO HDSS site has been capturing detailed longitudinal information related to health and diseases among individuals and households to improve the holistic experience in the local community and the wider population. SEACO remains the sole HDSS with International Standards Organisation (ISO) certification for quality data collection and data management processes (ISO 9001:2015) among over 48 HDSS from LMICs worldwide (http://www.indepth-network.org/). Besides the MUTUAL consortium, SEACO is also a member of the International Network for the Demographic Evaluation of Populations and Their Health network and the recently established HDSS-Asia network.

Comprehensive accounts of sociodemographic, lifestyle and medical information were gathered from all participants recruited to the population-wide baseline Core Census 2012 in an electronic handheld device-assisted face-to-face interview administered by trained interviewers. There were 45 246 community-dwelling residents aged 0–100 from 13 431 households enrolled in the Core Census 2012 for baseline enumeration. Of the participants, 38 228 participants were ≥18 years old (n=31 595) with a mean age of 45 years (SD 16.9 years). In 2013–2014, the first 5-yearly population-wide health survey, Health Round wave 1, was undertaken at the individual level among 25 168 participants aged 5 years and above who were part of the core sample. An overview of the population's overall health was assessed, including medical history, mental health, well-being, quality of life, lifestyle factors and health service utilisation. Of them, 18 101 adults were aged 18 and older (mean age 47.3 years (SD 16.3 years)). Data from this cohort of Health Round wave 1 (2013/2014) and the Core Census 2012 were included in the current study for a longer list of chronic

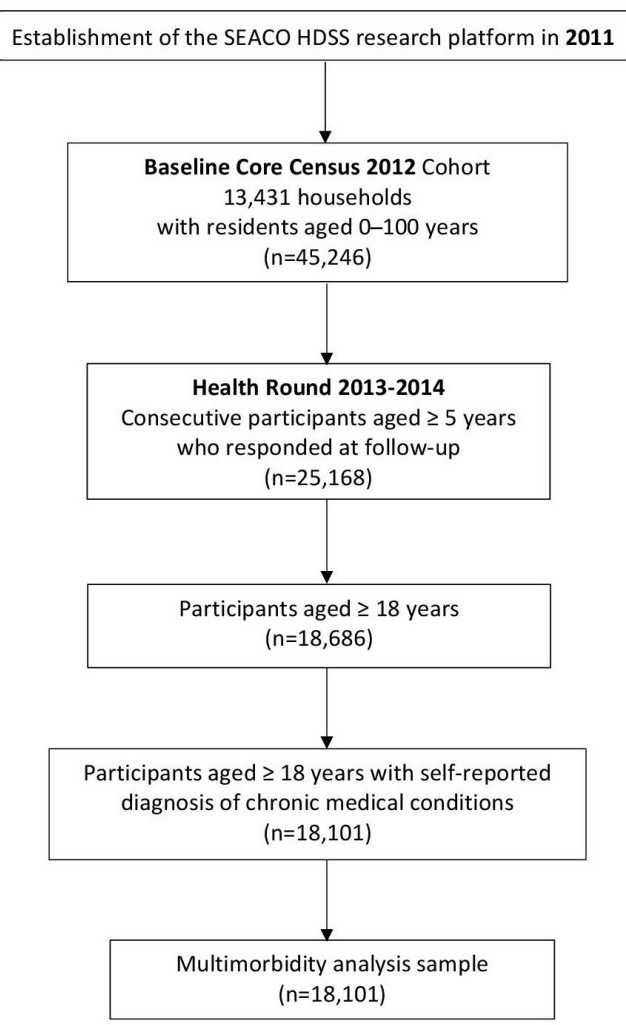

**Figure 1** Study sampling flow chart. SEACO, South East Asia Community Observatory; HDSS, Health and Demographic Surveillance System.

conditions to be considered. A flow chart of the population in this study is presented in figure 1.

## Chronic condition measures and definition of multimorbidity

While chronic health conditions were self-reported and not directly based on formal clinical diagnoses, participants were asked whether they had ever been told by medical professionals that they had a particular chronic health condition based on a series of dichotomous questions (yes/no). The conditions were heart disease (including coronary artery disease, myocardial infarction, angina pectoris, congestive heart failure), stroke, type 1 and/or type 2 diabetes mellitus (DM), hypertension, chronic kidney disease, end-stage renal failure (ESRF) requiring dialysis, arthritis, joint pain, back pain, asthma, vision problem, hearing problem and physical mobility problem.

Additionally, obesity (body mass index, BMI ≥30 kg/m$^2$) was included and defined using the principal BMI classification by the WHO,[19] by which weight (in kilograms) was divided by height squared (in metres). Weight and height were estimated using electronic portable weighing scales

with electronic height sensors (Patient Focus, model GBS-721), in which the participant was barefoot and wearing light clothing during the measurement. Considering the heterogeneity in the multimorbidity definitions is always high, and obesity was not consistently included in the list of chronic conditions for multimorbidity despite being a chronic and complex condition, these may potentially underestimate the prevalence of multimorbidity. Moreover, including obesity in multimorbidity estimates is crucial due to the well-studied connections between obesity and a range of complications, including three out of the ten leading causes of death worldwide—ischaemic heart disease, stroke and type 2 DM.[20] As such, we included obesity in the multimorbidity definition in our study.

Due to collinearity, participants who had any musculoskeletal disorder, either arthritis or joint pain or back pain or all three, were combined into one condition—'arthritis, joint pain and/or back pain'. Likewise, the mutually exclusive chronic kidney disease and ESRF requiring dialysis were considered together as 'chronic kidney disease'. To avoid potential bias caused by spurious associations, medical conditions with a prevalence of ≤1% were excluded, such as speech problem and gastrointestinal conditions. Acute infectious diseases consisting of dengue fever and urinary tract infection were also excluded. Each of the included 11 chronic conditions was coded as binary, where presence=1 and absence=0. In this study, multimorbidity is defined as the presence of two or more of these concurrent long-term physical conditions for each participant.

## Mental health measures

Mental health was measured using the Depression, Anxiety and Stress Scale-21 (DASS-21),[21] the short form of the DASS-42.[22] In the SEACO HDSS site, the Malay version of DASS-21 previously translated and validated in Malaysia[23] was adopted. It is a set of three self-report scales designed to measure the emotional states of depression, anxiety and stress. Each of the three DASS-21 scales contains seven items that are divided into subscales with similar content. The subscales correspond to the following diagnostic categories: 'depression' for mood disorders, 'anxiety' for panic disorder and 'stress' for generalised anxiety disorder. The individual scale items were scored on a 4-point Likert scale (never, sometimes, often and almost always) of frequency or severity of participants' experiences over the past week. The score of each subscale was summed and multiplied by two to be comparable to the long form of DASS. Higher scores indicate greater levels of symptoms. Participants with a cut-off score of ≥10 were classified as having depressive symptoms.[22]

## Study variables

Sociodemographic variables such as age (18–24, 25–34, 35–44, 45–54, 55–64, ≥65), sex (female or male), ethnicity (Malay, Chinese, Indian, other), education attainment

(no formal education, primary, secondary, tertiary education), employment status (working/self-employed, homemaker, unemployed, retired/pension, student), personal monthly gross income (<MYR1000, MYR1000–MYR1999, ≥MYR2000), monthly household income (grouped into bottom 40% (B40), middle 40% (M40), top 20% (T20)) and marital status (categorised as married, never married, widowed/divorced/separated) were included. Private health insurance was coded into two categories (yes through self-subscription/employer or no), and payer for health cost referred to either self/family member, government/pension, employer-provided health cover or personal health insurance. Health service utilisation included medication prescriptions in the last 2 weeks (ie, antidiabetic and antihypertensive drugs, coded into yes or no), hospital admissions (≥1 night in the past 6 months, categorised as yes or no) and emergency department visits in the past 6 months (yes or no).

## Data analysis
Descriptive statistics were used to summarise information regarding sociodemographic variables, clinical profiles, health service utilisation, and the prevalence of multimorbidity and individual chronic diseases. The age groups were compared using one-way analysis of variance (ANOVA) with post hoc Bonferroni test or $\chi^2$ test. The multimorbidity groups (yes or no) were compared using independent t-test or $\chi^2$ test. The total number of chronic conditions was counted and categorised as no multimorbidity (0–1) and multimorbidity (2, 3 and ≥4 chronic conditions, respectively). A multivariable logistic regression model using the enter method with entry criteria of α=0.05 and removal criteria=0.10 was performed to identify factors associated with multimorbidity (yes/no), controlled for mental health (depressive symptoms) and sociodemographic covariates (age, sex, ethnicity, education level and employment status). Covariates were selected, further categorised and included based on previous literature and theory. Diagnostic tests for complete case models were conducted. All analyses were performed using Stata/MP (V.17, StataCorp) and IBM SPSS Statistics (V.28, IBM). All statistical analyses were two sided with a significance of p<0.05.

## Patient and public involvement
None.

## RESULTS
Socioeconomic and demographic characteristics of the study cohort are reported in table 1. The average (range) age of the participants was 47 (18–97) years, 55.6% were females, two-thirds were Malays and the majority were married. The study population largely drawn from a semirural state in Malaysia was mainly represented by subjects who had lower education, and household and personal income levels (table 1).

Multimorbidity was identified in 28.5% of participants, with 15.9% having two conditions, and 12.6% having three or more conditions. See table 2 for the clinical characteristics of the study cohort, including the prevalence of multimorbidity and individual chronic conditions. Approximately 44.3% of the cohort had no chronic condition, and 27.2% had one condition. The individual prevalence of the chronic conditions ranged from 1.0% to 24.7% (range 0–13), with musculoskeletal disorder (24.7%) as the most prevalent chronic condition, followed by obesity (20.7%) and hypertension (18.4%). More than half of those with a chronic condition lived with multimorbidity. Additionally, depressive symptoms (not included in multimorbidity count) were observed to be affecting 17.6% of the study population. The prevalence of multimorbidity also increased steadily with age, from 9.6% among those younger than 35 years to as high as 46.8% in the elderly group (table 2). The average number of chronic conditions tripled from the youngest adult group to the oldest age group. The prevalence of obesity and overweight was observed to be at an alarming stage, as was present in nearly 40% of the young adults (18–34 years), 62.8% of the middle-aged group (35–59 years) and over half (54.7%) of the elderly population (aged 60 and above) (table 2).

Figure 2 displays the graphical trends of the number of chronic conditions and multimorbidity of the study population by age groups (18–24, 25–34, 35–44, 45–54, 55–64, ≥65 years). The number of chronic conditions and multimorbidity increased linearly with age (p<0.001). The prevalence of any chronic condition increased from 27.9% in the youngest age group (18–24 years) to as high as 55.7% in the group of ≥65 years (figure 2). Furthermore, multimorbidity and number of chronic conditions were observed to be more prevalent for individuals of female sex (p<0.001), except for participants with ≥6 chronic conditions, as demonstrated in table 1 and figure 3.

The results of multivariable logistic regression analysis assessing the association between sociodemographics, mental health and multimorbidity are shown in table 3. Older age was associated with a higher likelihood of having multimorbidity (adjusted OR 1.05, 95% CI 1.05 to 1.05).

In addition to older age, results from the multivariable logistic regression show that multimorbidity was associated with other sociodemographic factors, such as female sex, education level and employment status (table 3). Specifically, women are 1.28 times (95% CI 1.17 to 1.40) more likely to experience multimorbidity compared with men. Similarly, having no formal education increased the odds of developing multimorbidity (primary education: adjusted OR 0.63, 95% CI 0.53 to 0.74, secondary education: adjusted OR 0.60, 95% CI 0.51 to 0.70; tertiary education: adjusted OR 0.65, 95% CI 0.54 to 0.80). Working group was less likely to develop multimorbidity than retirees (adjusted OR 0.70, 95% CI 0.60 to 0.82).

**Table 1** Summary of population characteristics, showing sociodemographic characteristics (overall and by multimorbidity status; n=18 101)

| | | Multimorbidity | | P value |
| | Overall (n=18 101) | Yes (n=4669) (28.5%) | No (n=11 688) (71.5%) | |
|---|---|---|---|---|
| Age (years) | | | | <0.001 |
| Mean±SD | 47.3±16.3 | 55.3±13.8 | 44.0±15.8 | |
| Sex, n (%) | | | | <0.001 |
| Female | 9089 (55.6%) | 2824 (60.5%) | 6265 (53.6%) | |
| Male | 7268 (44.4%) | 1845 (39.5%) | 5423 (46.4%) | |
| Ethnicity, n (%) | | | | 0.005 |
| Malay | 10 741 (65.7%) | 3025 (64.9%) | 7716 (66.1%) | |
| Chinese | 3586 (21.9%) | 1100 (23.6%) | 2486 (21.3%) | |
| Indian | 1594 (9.8%) | 419 (9.0%) | 1175 (10.1%) | |
| Other | 418 (2.6%) | 119 (2.6%) | 299 (2.6%) | |
| Marital status, n (%) | | | | <0.001 |
| Never married | 2584 (16.0%) | 304 (6.5%) | 2280 (19.8%) | |
| Married | 11 846 (73.2%) | 3568 (76.7%) | 8278 (71.8%) | |
| Widowed/divorced/separated | 1745 (10.8%) | 781 (16.8%) | 964 (8.4%) | |
| Education level, n (%) | | | | <0.001 |
| No formal education | 841 (5.2%) | 314 (6.9%) | 527 (4.6%) | |
| Primary | 4842 (30.1%) | 1324 (28.9%) | 3518 (30.6%) | |
| Secondary | 9012 (56.1%) | 2529 (55.2%) | 6483 (56.4%) | |
| Tertiary | 1373 (8.5%) | 411 (9.0%) | 962 (8.4%) | |
| Employment status, n (%) | | | | <0.001 |
| Working/self-employed | 8189 (50.3%) | 1749 (37.6%) | 6440 (55.4%) | |
| Homemaker | 5017 (30.8%) | 1704 (36.7%) | 3313 (28.5%) | |
| Unemployed | 1868 (11.5%) | 743 (16.0%) | 1125 (9.7%) | |
| Retired/pension | 885 (5.4%) | 421 (9.1%) | 464 (4.0%) | |
| Student | 313 (1.9%) | 32 (0.7%) | 281 (2.4%) | |
| Personal gross monthly income (MYR) | | | | <0.001 |
| Mean±SD | 1204.0±1515.4 | 1094.7±1684.1 | 1224.0±1383.4 | |
| <1000 | 8554 (52.3%) | 2707 (58.0%) | 5847 (50.0%) | |
| 1000–1999 | 4575 (28.0%) | 1107 (23.7%) | 3468 (29.7%) | |
| ≥2000 | 3228 (19.7%) | 855 (18.3%) | 2373 (20.3%) | |
| Household monthly income (MYR)* | | | | <0.001 |
| Mean±SD | 3193.8±3537.1 | 2720.9±3333.0 | 3224.2±3357.7 | |
| Bottom 40% (B40) | 12 263 (75.0%) | 3679 (78.8%) | 8584 (73.4%) | |
| Middle 40% (M40) | 3076 (18.8%) | 749 (16.0%) | 2327 (19.9%) | |
| Top 20% (T20) | 1018 (6.2%) | 241 (5.2%) | 777 (6.6%) | |
| Private health insurance (through employer/self-subscription), n (%) | | | | <0.001 |
| Yes | 3650 (22.6%) | 955 (20.7%) | 2695 (23.3%) | |
| No | 12 507 (77.4%) | 3652 (79.3%) | 8855 (76.7%) | |

**Table 1** Continued

| | Overall (n=18 101) | Multimorbidity | | P value |
| | | Yes (n=4669) (28.5%) | No (n=11 688) (71.5%) | |
|---|---|---|---|---|
| Payer for healthcare cost, n (%) | | | | <0.001 |
| Self/family member | 13 329 (82.0%) | 3715 (80.1%) | 9614 (82.8%) | |
| Government/pension | 2293 (14.1%) | 805 (17.4%) | 1488 (12.8%) | |
| Employer-provided health cover | 492 (3.0%) | 85 (1.8%) | 407 (3.5%) | |
| Personal private health insurance | 139 (0.9%) | 32 (0.7%) | 107 (0.9%) | |

Numbers may not add up to totals due to missing data.
Statistical significance testing by $\chi^2$ test and independent t-test.
MYR1 was equivalent to the British Pound (GBP) £0.18 at the time of publication.
*Categorised as three income thresholds according to the local Department of Statistics, where B40, M40 and T20 represent groups with monthly income of <MYR3860, MYR3860–MYR8319 and >MYR 8319, respectively.
MYR, Malaysian ringgit.

## DISCUSSION

This population-based study provided evidence on the magnitude of multimorbidity and individual chronic conditions in a HDSS-scale community-dwelling general adult population. In addition, we identified various sociodemographic factors that could play a significant role in contributing to multimorbidity.

### Prevalence of multimorbidity and individual chronic conditions

Using simple disease counts of 11 selected chronic conditions, we estimated the prevalence of multimorbidity at 28.5% in the overall study population. Musculoskeletal disorder, obesity, hypertension and depressive symptoms are the most prominent physical and mental health conditions in this population. As expected, multimorbidity is more prevalent in older adults, affecting nearly half of the individuals aged 65 years and older (46.8%). However, it is worth noting that multimorbidity is affecting almost 10% of the young adults (18–34 years) and approximately one-third of the middle-aged adults (35–59 years), implying that multimorbidity is not just an issue of the elderly.

Extensive global and regional research has been conducted, showing substantial variability in the prevalence of multimorbidity. A systematic review and meta-analysis of 68 community-based studies worldwide by Nguyen et al[8] reported a prevalence range of multimorbidity of 3.5%–70% in 37 HICs and 1%–90% in 31 LMICs.[8] However, the number of diseases used as cut-off points for disease counts, disease combination, age ranges and definitions of multimorbidity differ widely across the studies. These inevitable heterogeneities, subsequently, generated inconsistent measures and prevalence of multimorbidity that hindered the comparability of findings between studies. Acceptance of a standard methodology will provide more clarity on the epidemiology of multimorbidity. The prevalence of multimorbidity in other South East Asian countries is very limited, and varies ethnically and geographically. An Indonesian community-based survey conducted among 9438 adults aged 40 years and above found a higher prevalence of multimorbidity (35.7%) than that of our findings. Nevertheless, the prevalence of multimorbidity in our studies is similar but higher than in Singapore (26.2%)[24] as well as the few local studies in Malaysia,[12 13 15–17] including the only three studies conducted by in-country researchers.[15–17] The measures of these studies were constrained by the number of considered chronic conditions (ie, <10 diseases), relatively old data (ie, 20 years ago) or only involved older cohorts (aged ≥56 years). Therefore, this study has addressed recent calls for investigations in general and multiethnic population samples, assessing a broader range of chronic conditions characterised by real-world community-based large data comprising medical diagnoses, anthropometric measurements, psychological and environmental risk factors.

### Factors associated with multimorbidity

Our findings are in agreement with the known paradox and other studies that multimorbidity increases with age, as long-term diseases accumulate with age.[10 25] It is interesting to note that in our study population, older age only explained part of the augmented risk of multimorbidity. We identified that other factors (sex, education level and employment status) are responsible for a substantial part of the association with multimorbidity. The inconsistency in sex differences and multimorbidity is not new, although most studies reported a higher prevalence of multimorbidity in women.[26 27] In this study, women were also more likely to be affected by multimorbidity. The findings are congruent with the literature in LMICs,[15 27 28] but not with the findings of the Collaborative Research on Ageing in Europe study conducted in Finland, Poland and Spain,[26] the HICs. This discrepancy could be attributed to shared effects of biological, social, psychological and environmental factors that generated inequalities in health status.[29 30] Education level appears

 Tan MMC, *et al. BMJ Open* 2022;**12**:e068172. doi:10.1136/bmjopen-2022-068172

**Table 2** Clinical characteristics of the study cohort (overall and stratified by age group; n=18 101)

| | Overall (n=18 101) | Age groups (years) | | | P value |
|---|---|---|---|---|---|
| | | 18–34 (n=4654) (25.7%) | 35–59 (n=9029) (49.9%) | 60 and above (n=4418) (24.4%) | |
| **Chronic conditions, n (%)** | | | | | <0.001 |
| Heart disease | 553 (3.2%) | 46 (1.0%) | 207 (2.4%) | 300 (7.0%) | |
| Stroke | 208 (1.2%) | 7 (0.2%) | 87 (1.0%) | 114 (2.6%) | |
| Diabetes mellitus* | 1868 (10.3%) | 44 (0.9%) | 975 (10.8%) | 849 (19.2%) | |
| Hypertension* | 3325 (18.4%) | 75 (1.6%) | 1633 (18.1%) | 1617 (36.6%) | |
| Chronic kidney disease | 173 (1.0%) | 17 (0.4%) | 68 (0.8%) | 88 (2.0%) | |
| Arthritis, joint pain and/or back pain | 4296 (24.7%) | 509 (11.5%) | 2033 (23.4%) | 1754 (40.7%) | |
| Asthma | 666 (3.8%) | 123 (2.8%) | 310 (3.5%) | 233 (5.4%) | |
| Obesity | 3665 (20.7%) | 712 (15.4%) | 2202 (24.7%) | 751 (18.0%) | |
| Vision problem | 2247 (12.4%) | 221 (4.7%) | 1182 (13.1%) | 844 (19.1%) | |
| Hearing problem | 489 (2.7%) | 75 (1.6%) | 154 (1.7%) | 260 (5.9%) | |
| Physical mobility problem | 1032 (5.7%) | 238 (5.1%) | 481 (5.3%) | 313 (7.1%) | |
| **Multimorbidity** | | | | | |
| Mean no of conditions±SD n (%) | 1.0±1.2 | 0.5±0.8 | 1.1±1.2 | 1.6±1.4 | <0.001 |
| No multimorbidity | 11 688 (71.5%) | 3777 (90.4%) | 5803 (70.6%) | 2108 (53.2%) | <0.001 |
| Zero condition | 7237 (44.3%) | 2815 (67.4%) | 3412 (41.5%) | 1010 (25.5%) | |
| One condition | 4449 (27.2%) | 962 (23.0%) | 2389 (29.1%) | 1098 (27.7%) | |
| With multimorbidity | 4669 (28.5%) | 401 (9.6%) | 2413 (29.4%) | 1855 (46.8%) | |
| Two conditions | 2595 (15.9%) | 287 (6.9%) | 1424 (17.3%) | 884 (22.3%) | |
| Three conditions | 1278 (7.8%) | 92 (2.2%) | 627 (7.6%) | 559 (14.1%) | |
| ≥Four conditions | 791 (4.8%) | 21 (0.5%) | 360 (4.4%) | 410 (10.4%) | |
| **BMI categories** | | | | | <0.001 |
| Obesity (BMI ≥30.0 kg/m$^2$) | 3665 (20.7%) | 712 (15.4%) | 2202 (24.7%) | 751 (18.0%) | |
| Overweight (BMI 25.0–29.9 kg/m$^2$) | 5994 (33.8%) | 1063 (23.0%) | 3396 (38.1%) | 1535 (36.7%) | |
| Normal (BMI 18.5–24.9 kg/m$^2$) | 7309 (41.3%) | 2452 (53.1%) | 3132 (35.1%) | 1725 (41.3%) | |
| Underweight (BMI<18.5 kg/m$^2$) | 746 (4.2%) | 390 (8.4%) | 186 (2.1%) | 170 (4.1%) | |
| **Depressive symptoms, n (%)** | | | | | <0.001 |
| Yes | 3138 (17.6%) | 891 (19.6%) | 1570 (17.7%) | 677 (15.6%) | |
| **Medication prescription, n (%)** | | | | | |
| Antidiabetic medication | 1023/1537 (66.6%) | 10 (23.8%) | 531 (66.0%) | 482 (69.9%) | <0.001 |
| Insulin | 425/1537 (27.7%) | 4 (9.5%) | 232 (28.8%) | 189 (27.4%) | 0.024 |
| Oral hypoglycaemic agents | 920/1540 (59.7%) | 8 (19.0%) | 479 (59.4%) | 433 (62.6%) | <0.001 |
| Antihypertensive therapy | 1871/2847 (65.7%) | 19 (27.9%) | 857 (62.6%) | 995 (70.6%) | <0.001 |
| **Hospitalisation (≥1 night in the past 6 months)†, n (%)** | | | | | NA |
| Yes | – | – | – | 320/4414 (7.0%) | |
| **Emergency department visit (in the past 6 months)†, n (%)** | | | | | NA |
| Yes | – | – | – | 324/4414 (4.8%) | |

Numbers may not add up to totals due to missing data.
Statistical significance testing by $\chi^2$ test and one-way ANOVA with post hoc Bonferroni test.
*Self-reported diagnosis or medication use.
†Only participants aged 60 years and above.
ANOVA, analysis of variance; BMI, body mass index; NA, not applicable.

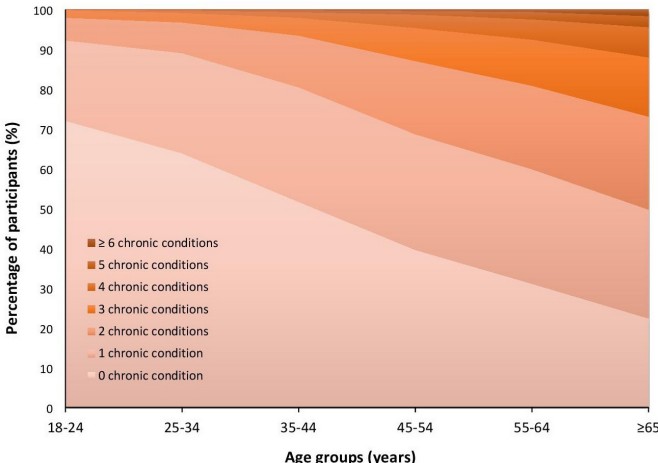

**Figure 2** Graphical representation of the prevalence of multimorbidity and number of chronic conditions of the study population by age groups (n=18 101).

to be associated with a higher risk of multimorbidity in our study. We found that individuals in general with no education had a higher likelihood of having multimorbidity than their counterparts. This is broadly in line with numerous other studies.[25 28 31 32] The impact of education on multimorbidity might be due to higher socioeconomic status and healthier behaviours and knowledge of individuals with higher education than those with less-educated backgrounds.[32] Employment status was associated with multimorbidity in the study population. Compared with retirees, working adults had a lower likelihood of developing multimorbidity. Age factor may potentially be underlying this relationship; this is not impossible when retirees are generally older than working individuals. Although highly prevalent among the participants, depressive symptoms did not appear to be a risk factor for multimorbidity in the study cohort. Contrarily, a systematic review suggested that the risk for depression was twice as great for people with multimorbidity compared with those without multimorbidity.[33] The cross-sectional nature of our data prevents us from establishing the

causal relationship between mental health and multimorbidity. This warrants further exploration in prospective cohort studies.

## Strengths and limitations

The strengths of this study include the use of a population-based dataset with high participation and completion rate (71.1%) that includes individuals of a broad range of ages, including those where chronic disease load begins to manifest. We have also captured a more comprehensive range of current and chronic conditions of ageing that have not been included in previous studies of multimorbidity. To the best of our knowledge, this is the first study investigating multimorbidity in a HDSS-scale real-world community-based general cohort in Malaysia. The regular surveillance of the large geographically defined population allowed for unique and detailed insight into epidemiological and health services changes. This includes the effects on individuals, households and entire communities. The breadth of sophisticated measures available for use as covariates, spanning socioeconomic and demographic domains, is a further strength of the current study, as it allowed a more thorough investigation of factors associated with multimorbidity than is usually attainable with medical records.

Nonetheless, the current research is not without limitations. First, causality cannot be assessed based on the nature of chronic condition data and hence cross-sectional study design. Future longitudinal investigation would allow an opportunity for causal inference. Second, the clinical diagnoses of medical conditions were self-reported by the participants, thus, may be susceptible to bias and inaccuracies that deviate from medical records. Nevertheless, self-report of diagnosis of chronic conditions has been widely used in epidemiological studies and proven valid.[34] Third, the reports of chronic conditions were restricted to the list included in the interview. As such, the prevalence of multimorbidity reported may be underestimated. However, we have selected the survey rounds with the longest lists of chronic conditions possible in our longitudinal databases and have included all relevant diseases for the assessment. Thus, we provided a detailed insight into multimorbidity in the study population. Fourth, given dementia is prevalent among older adults and people in this age group accounted for approximately 25% of the study sample, engaging in data collection of formal diagnoses of dementia is important and shall be considered in the definition of multimorbidity in the future. Lastly, it should be acknowledged that the surveys were conducted on participants who resided in semiurban and rural areas. In this regard, certain groups of the population may be under-represented, such as those with higher socioeconomic status, and patients who used private healthcare facilities centred around urban areas with more optimal disease control.

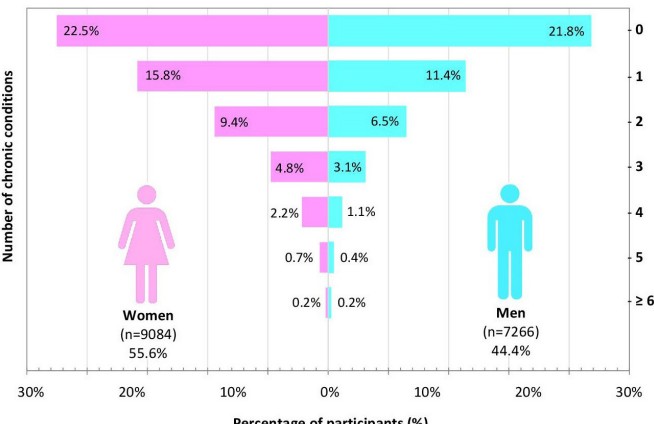

**Figure 3** Graphical representation of the sex-disaggregated prevalence of multimorbidity and number of chronic conditions of the study population.

**Table 3** Multivariable logistic regression modelling results of factors associated with multimorbidity (yes/no)

| Parameter | Crude OR | Adjusted OR | 95% CI | | P value for adjusted OR |
| --- | --- | --- | --- | --- | --- |
| | | | Lower bound | Upper bound | |
| Age | 1.05*** | 1.05*** | 1.05 | 1.05 | <0.001 |
| Sex | | | | | <0.001 |
| Male (Ref) | | 1 | | | |
| Female | 1.33*** | 1.28*** | 1.17 | 1.40 | |
| Ethnicity | | | | | 0.356 |
| Malay (Ref) | | 1 | | | |
| Non-Malay | 1.06 | 1.04 | 0.96 | 1.12 | |
| Education level | | | | | <0.001 |
| No formal education (Ref) | | 1 | | | |
| Primary | 0.63*** | 0.63*** | 0.53 | 0.74 | <0.001 |
| Secondary | 0.66*** | 0.60*** | 0.51 | 0.70 | <0.001 |
| Tertiary | 0.72*** | 0.65*** | 0.54 | 0.80 | <0.001 |
| Employment status | | | | | <0.001 |
| Working/self-employed | 0.30*** | 0.70*** | 0.60 | 0.82 | <0.001 |
| Student/homemaker/unemployed | 0.60*** | 0.87 | 0.74 | 1.02 | 0.077 |
| Retiree (Ref) | | 1 | | | |
| Mental health | | | | | 0.816 |
| No depressive symptoms (Ref) | | 1 | | | |
| Depressive symptoms | 0.96 | 1.01 | 0.91 | 1.13 | |

Diagnostic tests for complete case models: Nagelkerke $R^2$=0.159.
*p<0.05, **p<0.01, ***p<0.001.
Ref, Reference group.

### Implications for policy, practice and future research

The knowledge and evidence deficiencies of the epidemiology of multimorbidity in Malaysia are now addressed. They are vital steps towards future research and sensible governance and policy development to prevent developing multimorbidity and tackle the growing problems multimorbidity brings. Our findings facilitate the identification of high-risk populations and suggest a scope for a prevention strategy focusing on reducing the development of multimorbidity. Given the high prevalence of multimorbidity and the expected increase in the future, the current single-disease-specific design of the health system should also be accompanied by strategies to address complexities associated with multimorbidity. Our findings suggest that the key to managing multimorbidity may be to strengthen a realistic and multidisciplinary approach that simultaneously targets health outcomes and socioeconomic and demographics. The findings of this study could also be applicable for improving health outcomes and policy-making in other countries with social and economic developments comparable to that in Malaysia. We propose further research on the efficacy and effectiveness of trials is critical to be tested and established in better-equipped scientific settings with trained research staff. Therefore, funding and initiatives are important to translate the research findings into real-world settings, while exploring cost-effectiveness as a dual strategy in a resource-limited setting. Identifying specific clusters of chronic diseases that could cause the most significant problems is beyond the scope of this study, although its importance. Further research on multimorbidity clusters alongside their risk factors is needed to clarify relevant mechanisms to facilitate the development of targeted interventions for patients.

### CONCLUSION

The study shows that multimorbidity is common among adults of any age. The prevalence and risk of multimorbidity increased with age. Multimorbidity is also associated with sex, education level and employment status, indicating that multimorbidity is not just an issue of the elderly. Besides contributing to the developing literature for multimorbidity in LMICs, this study highlights the importance of prioritising health policies, integrated care interventions and clinical practice. All these should be oriented to implementing and delivering changes that include evidence-based components for effective preventative intervention and management of multimorbidity.

## Author affiliations
[1]Department of Health Service and Population Research, Institute of Psychiatry, Psychology and Neuroscience, King's College London, London, UK
[2]Victorian Heart Institute, Faculty of Medicine, Nursing and Health Sciences, Monash University, Clayton, Victoria, Australia
[3]Edinburgh Dementia Prevention, University of Edinburgh and Western General Hospital, Scotland, UK
[4]Department of Social Medicine, Heritage College of Osteopathic Medicine, Ohio University, Athens, Ohio, USA
[5]Global Public Health, Jeffrey Cheah School of Medicine and Health Sciences, Monash University Malaysia, Sunway City, Selangor, Malaysia
[6]South East Asia Community Observatory (SEACO), Jeffrey Cheah School of Medicine and Health Sciences, Monash University Malaysia, Sunway City, Selangor, Malaysia
[7]Centre for Innovative Drug Development and Therapeutic Trials for Africa (CDT-Africa), College of Health Sciences, Addis Ababa University, Addis Ababa, Ethiopia
[8]Department of Economics, College of Business and Economics, Jimma University, Jimma, Ethiopia
[9]Departamento de Psiquiatria e Psicologia Médica, Escola Paulista de Medicina, Universidade Federal de São Paulo, São Paulo, Brazil
[10]District Health Office Segamat, Ministry of Health Malaysia, Segamat, Johor, Malaysia
[11]Department of Sexual and Reproductive Health and Research, World Health Organization (WHO), Geneva, Switzerland
[12]International Institute for Global Health, United Nations University, Cheras, Kuala Lumpur, Malaysia
[13]Jeffrey Cheah School of Medicine and Health Sciences, Monash University Malaysia, Sunway City, Selangor, Malaysia
[14]Institute for Global Health and Development, Queen Margaret University, Edinburgh, UK

**Correction notice** This article has been corrected since it was first published. The open access licence has been updated to CC BY.

**Contributors** Conceptualisation: MMCT, AMP, GMT, DM and TTS. Data curation: MMCT, RI. Formal analysis: MMCT. Funding acquisition: AMP, GMT, DM, TTS. Investigation: MMCT, AMP, GMT, PA, DR and TTS. Methodology: MMCT, AMP, GMT, DM, PA, DR, TTS. Project administration: MMCT. Resources: MMCT, AMP, GMT, ZK, PA, DR and TTS. Software: MMCT, AMP. Supervision: MMCT, AMP, GMT and TTS. Visualisation: MMCT. Writing-original draft preparation: MMCT. Writing-review and editing: MMCT, AMP, GMT, DM, RI, EA, AÁMK, PA, DR and TTS. AMP is the guarantor of the manuscript.

**Funding** This work and publication of this article were supported by the Medical Research Council (UK Research and Innovation) grant (MUTUAL grant, King's College London, Grant number MR/T037423/1 awarded to AMP and team). Funding for SEACO HDSS was provided by the research offices of Monash University in Australia and Malaysia (Faculty of Medicine, Nursing and Health Sciences; Jeffrey Cheah School of Medicine and Health Sciences; and Faculty of Arts).

**Competing interests** None declared.

**Patient and public involvement** Patients and/or the public were not involved in the design, or conduct, or reporting, or dissemination plans of this research.

**Patient consent for publication** Not applicable.

**Ethics approval** This study involves human participants and was approved by Monash University Human Research Ethics Committee (MUHREC CF11/3663 – 2011001930 and 2013-3837-3646). Participants gave informed consent to participate in the study before taking part.

**Provenance and peer review** Not commissioned; externally peer reviewed.

**Data availability statement** Data are available on reasonable request. Data are available on request from the corresponding authors, or by application to the SEACO, Jeffrey Cheah School of Medicine and Health Sciences, Monash University Malaysia.

**ORCID iDs**
Michelle M C Tan http://orcid.org/0000-0002-4806-7939
A Matthew Prina http://orcid.org/0000-0001-6698-3263
Graciela Muniz-Terrera http://orcid.org/0000-0002-4516-0337
Devi Mohan http://orcid.org/0000-0002-0898-2729
Tin Tin Su http://orcid.org/0000-0003-0387-6406

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
