## [Reviewer comments · BMJ Open]

ARTICLE DETAILS

TITLE (PROVISIONAL)	Prevalence of, and factors associated with, multimorbidity among 18,101 adults in the South East Asia Community Observatory Health and Demographic Surveillance System in Malaysia: a population-based, cross-sectional study of the MUTUAL consortium
AUTHORS	Tan, Michelle; Prina, A.Matthew; Muniz Terrera, Graciela; Mohan, Devi; Ismail, Roshidi; Assefa, Esubalew; Keinert, Ana; bin Kassim, Zaid; Allotey, P; Reidpath, Daniel; Su, Tin

VERSION 1 – REVIEW

REVIEWER	Tran To Tran Nguyen University of Medicine and Pharmacy Ho Chi Minh City Hospital, Geriatrics
REVIEW RETURNED	29-Sep-2022

GENERAL COMMENTS	Thank you the authors for addressing multimorbidity in a general adult community. Please see below my comments to enhance your manuscript: * Introduction: 1. In the first paragraph, the author needs to add some evidence to clarify the statement “Multimorbidity is associated with lower quality of life, impaired functional capacity, psychological distress, increased premature mortality, and greater healthcare service utilisation and medical expenditure”2. Lines 136-138: the author wrote “it was found that the prevalence of multimorbidity is comparable between high-income countries (HICs) and low- and middle-income countries (LMICs)”. I think you should provide some numbers to illustrate the statement *Methods: 1. Lines 173-174: The data were drawn from SEACO HDSS database. Could the author give some more information about the participants in your study? For example: What areas were included in this study? (urban or rural?)2. Lines 203-206: “Chronic health conditions were self-reportedbased on a series of dichotomous questions (yes/no)”. Were the participants interviewed face to face or via phone? And how did you collect information in case participants had dementia?3. Dementia is prevalent in older people. Why did the author include this in multimorbidity while older adults accounted for approximately 25% of the participants?4. Lines 227-228: DASS-21 was used to evaluate participants’ mental health. Was it translated and validated in Malaysia? * Results: Muscular skeletal disorder was the most prevalent condition in your study but the authors did not mention that how long a pain lasted was considered a muscular skeletal disorder in the Method part * Discussion:
--

	Lines 385-386: the authors should add references to support the statement.
--	--

REVIEWER	Rifqah Roomaney South African Medical Research Council, Burden of Disease Research Unit, Tygerberg
REVIEW RETURNED	31-Oct-2022

GENERAL COMMENTS	Article: Prevalence and factors associated with multimorbidity in 18,101 adults in Malaysia: 3 A population-based study of the MUTUAL consortium Manuscript number: bmjopen-2022-068172 Thank you for the opportunity to review this excellent manuscript. In general, it is well written, clear and interesting. Assessing the prevalence of multimorbidity is an important issue for health care planning and it is thus also topical. The study assesses multimorbidity in adults and not only older adults, which is indeed a gap in the literature, especially with regard to LMICs. The methods are solid and the interpretation of the results is good. I have a few minor suggestions / comments for the authors to consider. Methods  • Line 187, Page 6. Hyphen missing between 0 and 100. • Obesity is sometimes included as a risk factor in studies of multimorbidity and other times included as a disease condition. Can the authors elaborate upon this in either the discussion, methods or limitations? What would excluding it do? Results  • Line 298, Page 12: Were depressive symptoms not included the conceptualization of multimorbidity? Please make this clear in the methods. If it is not included in the concept of multimorbidity and not as a risk factor (in the logistic regression), why is it reported on? • For Table 3, can stars (asterisks) also be added for the Adjusted ORs? It's a bit confusing to only have the asterisks for the crude ORs. Discussion  • Can more detail be added on how the prevalence relate to studies in other LMICs in the region? Is it similar to what's found in other SEA regions? Conclusion  • Are there plans to analyse what the specific disease patterns among the multimorbid are? Supplementary  • I would suggest adding sex-disaggregated data on multimorbidity, as it is more common in women.
--

VERSION 1 – AUTHOR RESPONSE

Reviewer #1

- 1) *Thank you the authors for addressing multimorbidity in a general adult community. Please see below my comments to enhance your manuscript:
Introduction: In the first paragraph, the author needs to add some evidence to clarify the statement "Multimorbidity is associated with lower quality of life, impaired functional capacity, psychological distress, increased premature mortality, and greater healthcare service utilisation and medical expenditure".*

Our response:

Thank you for your kind feedback to enhance our paper. It is true that some evidence is needed to clarify the abovementioned statement. The references for the evidence were initially all cited at the end of the sentence. We have now cited each of the evidence for further clarity, as shown in the Introduction section on Page 4 of our revised manuscript (Lines 145-147).

- 2) *Introduction: Lines 136-138: the author wrote “it was found that the prevalence of multimorbidity is comparable between high-income countries (HICs) and low- and middle-income countries (LMICs)”. I think you should provide some numbers to illustrate the statement*

Our response:

We agree with you. The prevalence and 95% CIs are now added to the Introduction section of our revised paper, side-by-side with the countries in the statement, as demonstrated on Pages 4 and 5.

- 3) *Methods: Lines 173-174: The data were drawn from SEACO HDSS database. Could the author give some more information about the participants in your study? For example: What areas were included in this study? (urban or rural?)*

Our response:

We have now added more relevant information in the first paragraph of the Methods section, as shown on Page 6 of our revised manuscript. The relevant text reads: “Data for the snapshot analyses were drawn from the South East Asia Community Observatory (SEACO) HDSS database established in 2011 in semi-urban and rural areas within the Segamat district of Johor state, Malaysia”.

- 4) *Methods: Lines 203-206: “Chronic health conditions were self-reportedbased on a series of dichotomous questions (yes/no)”. Were the participants interviewed face to face or via phone? And how did you collect information in case participants had dementia? Dementia is prevalent in older people. Why did the author include this in multimorbidity while older adults accounted for approximately 25% of the participants?*

Our response:

Thank you for bringing up these valid discussion points. In the SEACO HDSS site, the interviews have been implemented as face-to-face interviews. For further clarity, we have now modified the term ‘personal interview’ to ‘face-to-face interview’, as shown on Page 7 of our revised manuscript. Participants with severe dementia were excluded from the Health Round interviews, thus, were not included in the present analysis. Whereas participants with mild dementia (assessed by subjective memory complaints) were accompanied by their families in the face-to-face interviews, thus, we did not make this a key issue in our current analysis. On the other hand, we also do understand and agree that dementia is prevalent among older people and that older adults accounted for approximately one-fourth of our study sample, thus, we have now acknowledged this as an important limitation of our study, as shown in the marked text on Page 20 of our revised paper. We sincerely intend to collect formal diagnoses of dementia among the participants in the future.

- 5) *Methods: Lines 227-228: DASS-21 was used to evaluate participants’ mental health. Was it translated and validated in Malaysia?*

Our response:

Yes, DASS-21 was translated and the Malay version was validated by Professor Ramli Musa (Musa R et al. 2007), and has been adopted by SEACO HDSS site. We have now clarified this in our revised paper, as shown in marked text on Page 9 in the Methods section.

Reference: Musa R, Fadzil MA, Zain Z. Translation, validation and psychometric properties of Bahasa Malaysia version of the Depression Anxiety and Stress Scales (DASS). ASEAN Journal of Psychiatry. 2007;8(2):82-9.

- 6) *Results: Muscular skeletal disorder was the most prevalent condition in your study but the authors did not mention that how long a pain lasted was considered a muscular skeletal disorder in the Method part.*

Our response:

It is true that musculoskeletal disorder was the most prevalent condition in our study. As explained in the Methods section, during the face-to-face interview, the participants were asked if they had ever been told by a medical doctor that he/she has chronic joint pain, back pain and/or arthritis, respectively. In other terms, the duration of pain was determined by physicians who made the clinical diagnoses of the musculoskeletal disorders. We hope this clarifies.

- 7) *Discussion: Lines 385-386: the authors should add references to support the statement.*

Our response:

References are now added to support the statement, as shown in the marked text of our revised paper (Page 18 and Reference list).

Reviewer #2

- 1) *Thank you for the opportunity to review this excellent manuscript. In general, it is well-written, clear and interesting. Assessing the prevalence of multimorbidity is an important issue for health care planning and it is thus also topical. The study assesses multimorbidity in adults and not only older adults, which is indeed a gap in the literature, especially with regard to LMICs. The methods are solid and the interpretation of the results is good. I have a few minor suggestions/comments for the authors to consider.*

Methods: Line 187, Page 6. Hyphen missing between 0 and 100.

Our response:

Thank you so much for your time and kind effort in evaluating our paper and for all of your kind feedback, truly appreciate it. We would also like to express our utmost appreciation to you for valuing our work and for bringing up many valid discussion points for improvement.

The hyphen between 0 and 100 (Methods section, previously Line 187, now Line 207 in the marked text) was indeed missing in the peer-reviewed proof on the journal submission system, although the hyphen can still be seen in our original manuscript draft. Our sincere apologies and thanks much for noticing this and pointing it out to us, we have now further done some formatting in our Microsoft Word document to prevent the recurrence of the same issue when we submit our revised manuscript to the journal submission system. Please do not hesitate to let us know if this happens to occur again, thank you.

- 2) *Methods: Obesity is sometimes included as a risk factor in studies of multimorbidity and other times included as a disease condition. Can the authors elaborate upon this in either the discussion, methods or limitations? What would excluding it do?*

Our response:

We thank you for your kind insights and helpful suggestions. We agree with you that obesity is indeed sometimes included as a risk factor in studies of multimorbidity, and other times as a disease condition like our study. Considering the heterogeneity in the multimorbidity definitions is always high, and obesity was not consistently included in the list of chronic conditions for multimorbidity despite being a chronic and complex condition, potentially underestimating the prevalence of multimorbidity. Thus, we included obesity in the multimorbidity definition in our study. In addition, we think that including obesity in multimorbidity estimates is important due to the well-studied links between obesity and a range of complications, including three out of the ten leading causes of death worldwide – ischaemic heart disease, stroke and type 2 diabetes mellitus. We are glad that obesity was considered in the multimorbidity definition in our study, particularly since the result depicted that obesity is a significant contributor to the trend of multimorbidity in our study sample (i.e., second most prevalent chronic condition), comprised of 20.7% of the study cohort. As per your kind suggestion, we have also now added some elaboration around obesity and multimorbidity definition in the marked text in the Methods section of our revised paper (Page 8, Lines 236-243).

- 3) *Results: Line 298, Page 12: Were depressive symptoms not included the conceptualization of multimorbidity? Please make this clear in the methods. If it is not included in the concept of multimorbidity and not as a risk factor (in the logistic regression), why is it reported on?*

Our response:

We thank you for raising this question. Given physical illnesses are always one of the contributing factors to mental illness, we assessed the general pattern of depressive symptoms among the participants using the DASS-21 instrument, rather than being considered as a disease entity in this study (Pages 10, 13 and 14, and Table 2, highlighted in yellow). We have now also articulated our inclusion of depressive symptoms as a potential risk factor in the logistic regression model to investigate any additional bidirectional association between mental distress and multimorbidity (Pages 10, 15 and 16, and Table 3). There is however no significant relationship found between the two variables in our study cohort. It is interesting to investigate the causality using longitudinal data in other settings in LMICs in the future which we have also spelled out in the marked text of our revised paper.

- 4) *Results: For Table 3, can stars (asterisks) also be added for the Adjusted ORs? It's a bit confusing to only have the asterisks for the crude ORs.*

Our response:

Thank you for letting us know how you feel and our sincere apologies for the confusion. We are absolutely delighted to add stars (asterisks) for the adjusted-ORs as well, and we have now done so, as seen throughout Table 3. The exact *p* value for each adjusted-OR was additionally outlined in the right column of Table 3. We hope this helps.

- 5) *Discussion: Can more detail be added on how the prevalence relate to studies in other LMICs in the region? Is it similar to what's found in other SEA regions?*

Our response:

We thank you for your kind suggestion and we agree with you that it is important to detail the prevalence of multimorbidity in other LMICs and South East Asia region. In our revised paper, we now added the relevant prevalence of multimorbidity in two sections of our revised paper, including the Introduction (Pages 4 and 5) and Discussions sections (Page 17).

6) *Conclusion: Are there plans to analyse what the specific disease patterns among the multimorbid are?*

Our response:

Yes, we do have the plans and are currently working on the next paper focussing on the specific disease patterns among chronic conditions, to fill in another key gap in the multimorbidity field of study. The idea is to first identify multimorbidity patterns, followed by exploring how these specific disease patterns are associated with (a) 10-year all-cause mortality; (b) risk factors; and (c) World Health Organization Quality of Life (WHOQoL). We will be more than happy to share the paper with you upon publication if you are interested in it, thank you very much.

7) *Supplementary: I would suggest adding sex-disaggregated data on multimorbidity, as it is more common in women.*

Our response:

Thank you very much for your kind suggestion. We have now added the following graphical data of the sex-disaggregated prevalence of multimorbidity and the number of chronic conditions in our revised paper (Figure 3).

Figure 3 Graphical representation of the sex-disaggregated prevalence of multimorbidity and number of chronic conditions of the study population.

VERSION 2 – REVIEW

REVIEWER	Tran To Tran Nguyen University of Medicine and Pharmacy Ho Chi Minh City Hospital, Geriatrics
REVIEW RETURNED	03-Dec-2022

GENERAL COMMENTS	Thank you corresponding author for your thorough responses to my comments. I have no more questions about the revised manuscript.
---

REVIEWER	Rifqah Roomaney South African Medical Research Council, Burden of Disease Research Unit, Tygerberg
REVIEW RETURNED	28-Nov-2022

GENERAL COMMENTS	I am happy with the changes made by the authors.
--